# Greenland during the last interglacial: the relative importance of insolation and oceanic changes

Rasmus A. Pedersen[1,2], Peter L. Langen[2], and Bo M. Vinther[1]

[1]Centre for Ice and Climate, Niels Bohr Institute, University of Copenhagen, Copenhagen, Denmark
[2]Climate and Arctic Research, Danish Meteorological Institute, Copenhagen, Denmark

*Correspondence to:* Rasmus A. Pedersen (anker@nbi.ku.dk)

**Abstract.** Insolation changes during the Eemian (the last interglacial period, 129–116,000 years before present) resulted in warmer than present conditions in the Arctic region. The NEEM ice core record suggests warming of 8±4 K in northwestern Greenland based on stable water isotopes. Here we use general circulation model experiments to investigate the causes of the Eemian warming in Greenland. Simulations of the atmospheric response to combinations of Eemian insolation and pre-industrial oceanic conditions and vice versa, are used to disentangle the impacts of the insolation change and the related changes in sea surface temperatures and sea ice conditions. The changed oceanic conditions cause warming throughout the year, prolonging the impact of the summertime insolation increase. Consequently, the oceanic conditions cause annual mean warming of 2 K at the NEEM site, whereas the insolation alone causes an insignificant change. Taking the precipitation changes into account, however, the insolation and oceanic changes cause more comparable increases in the precipitation-weighted temperature, implying that both contributions are important for the ice core record at the NEEM site. The simulated Eemian precipitation-weighted warming of 2.4 K at the NEEM site is low compared to the ice core reconstruction, partially due to missing feedbacks related to ice sheet changes and an extensive sea ice cover. Surface mass balance calculations with an energy balance model further indicate that the combination of temperature and precipitation anomalies leads to potential mass loss in the north and southwestern parts of the ice sheet. The oceanic conditions favor increased accumulation in the southeast, while the insolation appears to be the dominant cause of the expected ice sheet reduction. Consequently, the Eemian is not a suitable analogue for future ice sheet changes.

## 1 Introduction

The last interglacial, the Eemian, was characterized by higher than present temperatures in the Arctic region driven by increased summertime insolation at high northern latitudes (CAPE-Last Interglacial Project Members, 2006; Masson-Delmotte et al., 2013). The recent NEEM ice core from northwestern Greenland covers the last interglacial period and indicates substantial warming from 129–114 thousand years before present (ka) peaking at 8±4 K above the mean of the last millennium (NEEM community members, 2013). This temperature estimate is based on stable water isotopes, specifically $\delta^{18}$O, using the temperature–isotope relation from the present interglacial (Vinther et al., 2009). A recent, alternate reconstruction based on isotopic air composition ($\delta^{15}$N) from the same ice core yields a very similar estimate of 7–11 K, with 8 K as the most likely

estimate (Landais et al., 2016). General circulation models, however, generally simulate a much more limited warming (Braconnot et al., 2012; Lunt et al., 2013; Masson-Delmotte et al., 2013; Otto-Bliesner et al., 2013; Landais et al., 2016), motivating further investigation of the mechanisms behind the Eemian warming in Greenland.

During the Eemian, the global sea level was increased 6–9 m above present (Dutton and Lambeck, 2012; Dutton et al., 2015; Kopp et al., 2009), indicating a substantial reduction of the continental ice sheets. Several studies have presented ice sheet model reconstructions of the Greenland ice sheet (GrIS), but the reconstructions vary substantially in both magnitude and spatial distribution of the ice sheet changes. Regarding the magnitude, the ensemble of reconstructions suggests a likely range of sea level contribution from GrIS of 1.4–4.3 m (Masson-Delmotte et al., 2013). Spatially, the potential changes as suggested by the various models are retreat in southwest (Helsen et al., 2013), retreat in north- and southwest with a separate South Dome ice cap (Robinson et al., 2011), retreat in southwest and north-northeast (Born and Nisancioglu, 2012; Quiquet et al., 2013; Stone et al., 2013), and complete loss of the South Dome (Cuffey and Marshall, 2000; Lhomme et al., 2005). Only very few constraints exist that can be used to assess these reconstructions. The deep Greenland ice cores that contain Eemian ice are obvious fix-points. These are NEEM (77.45° N, 51.06° W; NEEM community members, 2013), NGRIP (75.10° N, 42.32° W; NGRIP members, 2004), GRIP (72.5° N, 37.3° W; GRIP members, 1993) and GISP2 (72.58° N, 38.48° W; Grootes et al., 1993). Additionally, basal parts of the Renland (71.3° N, 26.7° W) and Camp Century (77.2° N 61.1° W) ice cores contain ice from the Eemian (Johnsen et al., 2001). The DYE-3 ice core further south (65.2° N 43.8° W; Dansgaard et al., 1982) has distorted layers making it difficult to assess the deepest part of the core (Johnsen et al., 2001), but the basal part contains ice older than the Eemian (Willerslev et al., 2007). Ocean sediment cores further indicate the presence of ice in southern Greenland during the Eemian (Colville et al., 2011).

Due to the lapse rate (i.e. decreasing temperature with atmospheric height), elevation changes will impact the surface temperature on GrIS. The NEEM ice core temperature reconstruction has been corrected for this effect using the ice core air content which suggests an elevation increase of 45±350 m relative to the present ice sheet elevation (NEEM community members, 2013). Besides the direct impact of the elevation change, changes in the GrIS topography could also impact the large scale circulation (Hakuba et al., 2012; Lunt et al., 2004; Petersen et al., 2004), as well as surface energy balance on Greenland (Merz et al., 2014a) and the ice core record through changed precipitation patterns (Merz et al., 2014b).

The conversion from $\delta^{18}O$ to temperature may be a contributing factor to mismatches between model simulations and $\delta^{18}O$ temperature reconstructions: The $\delta^{18}O$–temperature relationship is sensitive to precipitation intermittency, evaporation conditions, and atmospheric transport, and thus varies spatially and historically (Jouzel et al., 1997; Masson-Delmotte et al., 2011). Hence, sea surface warming and reduced sea ice extent might affect the $\delta^{18}O$ record, as illustrated by isotope-enabled climate model simulations (Sime et al., 2013). The NEEM $\delta^{18}O$ temperature estimate is based on the average Holocene $\delta^{18}O$–temperature relationship from other central Greenland ice cores (Vinther et al., 2009), but the actual relationship might be different due to the shifted location or the climatic changes during the Eemian.

One important factor to consider when interpreting ice core records is the precipitation seasonality. The ice core record reflects the snow deposition on the surface, and only records climatic information during snowfall events (Steig et al., 1994). In this way, the precipitation seasonality creates a bias towards seasons with more snow deposition, and changes in the seasonality

may induce changes in the ice core record even with an unchanged temperature (Persson et al., 2011). As an example, model simulations indicate that the present day climate in northwestern Greenland is biased towards summer due to precipitation seasonality (Steen-Larsen et al., 2011). Thus, when comparing model simulations to ice core records it is useful to consider the precipitation-weighted temperature to obtain a fair comparison. The precipitation-weighted temperature ($T_{pw}$) can be calculated as:

$$T_{pw} = \frac{\sum_{j=1}^{N} T_j p_j}{\sum_{j=1}^{N} p_j} \tag{1}$$

where $N$ denotes the total number of time samples, and $T_j$ and $p_j$ is the temperature and precipitation during the $j$th time sample. In our study, the weighting is based on monthly means of near surface air temperature and total precipitation.

Using a series of general circulation model (GCM) experiments, we assess the Greenland climate during the Eemian. We investigate how the simulated changes could affect the GrIS surface mass balance and the ice core record, and compare the reconstructed and simulated temperatures. While the insolation change is the only forcing in our experiments, we further compare the direct impact of the insolation change and the indirect effect of retreating sea ice and increasing sea surface temperatures (SSTs). The direct and indirect impacts are separated using two hybrid experiments: one forced by Eemian insolation and fixed pre-industrial sea surface conditions (direct impact, "insolation-only") and one with pre-industrial insolation and Eemian sea surface conditions (indirect impact, "ocean-only"). The temperature change during the Eemian resembles that of future climate scenarios (e.g. Clark and Huybers, 2009), and our comparison could reveal whether the Eemian is an appropriate analogue for future climate change in Greenland; i.e. whether insolation or the ambient oceanic warming dominates the total response.

In the assessment of the Greenland climate, we also aim to investigate how the simulated Eemian climate could impact the ice sheet. The ice sheet response is a combined result of dynamics (ice flow) and surface mass balance changes (melt and accumulation). Here, we employ a detailed surface scheme to assess the surface mass balance. Again, the assessment of the relative importance of the insolation and sea surface warming will indicate whether Eemian ice sheet reconstructions are useful analogues for future ice sheet changes.

The experiments and the employed models are described in Sect. 2. Results are presented and discussed in Sect. 3, followed by conclusions in Sect. 4.

## 2 Methods

### 2.1 Model configuration

The model used for this study is the EC-Earth global climate model (Hazeleger et al., 2010, 2012) in the most recent version 3.1. We employ the atmosphere-only configuration based on the IFS atmospheric model (cycle 36r4; European Centre for Medium-Range Weather Forecasts, 2010) in a T159 spectral resolution with an associated Gaussian grid of roughly 1.125° × 1.125° horizontal resolution and 62 layers in the vertical. In order to allow paleoclimate simulation, the model has been expanded with an option to modify the insolation according to any given orbital configuration. The insolation is internally calculated following Berger (1978) using the same code modification as Muschitiello et al. (2015).

Compared to the widely used version 2.3, which was included in the latest IPCC assessment report (Flato et al., 2013), the new EC-Earth version includes updated versions of both the atmosphere and ocean models. Comparison of a present-day simulation to gridded observational data reveals an improved overall performance in version 3.1 compared to the previous version with a few remaining biases (cf. Davini et al., 2014). Relevant for this study, the comparison reveals a cold bias over most of Greenland and a too extensive Arctic sea ice cover.

The boundary conditions for our experiments (sea surface temperature and sea ice concentration) are obtained from two simulations with the fully-coupled EC-Earth system: an Eemian experiment forced with 125 ka conditions (i.e. insolation and greenhouse gas concentrations; GHGs) and a pre-industrial control experiment. The coupled experiments are described in detail in Pedersen et al. (2016b).

Due to the diverse ice sheet reconstructions, we have kept the ice sheets fixed at present day extents in all of our simulations. Vegetation is similarly kept at present day values. Combined with the fixed SST and sea ice conditions, this means that our experiments do not include any additional feedbacks from the ocean, vegetation or ice sheet geometry (as discussed in Pedersen et al., 2016b).

## 2.2 Surface mass balance calculations

To investigate the impacts of the simulated climate changes on the GrIS surface mass balance, we performed off-line calculations with the subsurface scheme of the HIRHAM5 regional climate model (updated from Langen et al., 2015). The subsurface model was here run on the Gaussian grid associated with the EC-Earth experiments and forced at 6 hour intervals with incoming shortwave and downward longwave radiation, latent and sensible heat fluxes, along with rain, snow and evaporation/sublimation taken directly from the EC-Earth output. The subsurface model was updated slightly compared to that described by Langen et al. (2015); most notably it employs 25 layers with a total depth of 70 m water equivalent and includes temperature- and pressure-dependent densification of snow and firn (following Vionnet et al., 2012). It accounts for heat diffusion, vertical water transport and refreezing. Each layer can hold liquid water corresponding to 2 % of the snow pore space volume and excess water percolates downward to the next layer. When a layer density exceeds the pore close off density (830 kg m$^{-3}$), water percolating down from above is added to a slush layer and runs off exponentially with an exponential time scale depending on surface slope (Lefebre et al., 2003; Zuo and Oerlemans, 1996). Until it runs off, the slush layer water is available for superimposed ice formation onto the underlying ice layer at a rate that assumes a linear temperature profile in that layer.

## 2.3 Experimental design

We have designed four experiments to investigate how the last interglacial insolation changes impacted the climatic conditions on Greenland (cf. Table 1). An experiment with Eemian (125 ka) conditions ("iL+oL", full Eemian experiment) is compared to a pre-industrial control climate state ("iP+oP"). The simulations are forced with GHGs and insolation from the respective periods along with prescribed sea surface temperatures (SST) and sea ice concentration (SIC) obtained from fully coupled model experiments with identical GHGs and insolation (the coupled simulations are described in Pedersen et al., 2016b).

Aiming to disentangle the direct impact of the insolation changes and the indirect impact of changed sea surface conditions (SST and SIC), we have designed two hybrid experiments: The first experiment is forced by Eemian insolation and pre-industrial sea surface conditions ("iL+oP", insolation-only) and the second is conversely forced by pre-industrial insolation and Eemian sea surface conditions ("iP+oL", ocean-only).

During the Eemian at 125 ka the Northern Hemisphere summer solstice occurs near perihelion and the obliquity is increased compared to present day. The changed orbit causes an insolation increase over Greenland during summer compensated by a decrease during autumn, i.e. an earlier onset of the polar night (cf. Fig. 1). The presented results are all following a fixed seasonal calendar. Compared to an alternative angular calendar based on astronomical positions, this changes the seasonal variation of the insolation. As the calendar we use is defined with a fixed vernal equinox, the largest difference between the two calendar definitions is found during northern hemisphere autumn. Joussaume and Braconnot (1997) illustrate that the negative anomaly at high northern latitudes (cf. Fig. 1) does not appear with angular season definitions. Note, however, that annual mean anomalies are not affected by the choice of calendar.

In the coupled simulations from Pedersen et al. (2016b), the induced insolation forcing leads to sea ice retreat and increasing SSTs across high northern latitudes. Figure 2 depicts sea ice concentration and SST anomalies in the coupled simulations from Pedersen et al. (2016b), indicating the differences between the sea surface boundary conditions employed here. The sea ice reduction is primarily manifested as a northward retreat of the ice edge; the sea ice concentration in the central Arctic is largely unchanged. The strongest warming is found in the North Atlantic following the northward retreat of the sea ice edge and a strengthening of the Atlantic meridional overturning circulation (AMOC). The AMOC increase is related to a regional increase of the surface salinity and increased wintertime convection. The simulated pre-industrial climate has an extensive Arctic sea ice cover and a related lack of deep convection in the Labrador Sea. This contributes further to the Eemian North Atlantic warming which, despite regional agreement, is larger than suggested by proxy reconstructions in the central North Atlantic (see detailed description in Pedersen et al., 2016b).

All simulations have a total length of 60 years of which the first 10 are disregarded as spin-up. Statistical significance of changes is assessed using a two-sided Student's $t$ test (von Storch and Zwiers, 2001), taking into account the serial autocorrelation of the time series.

## 3   Results and Discussion

As described by Pedersen et al. (2016b), the anomalies between Eemian and pre-industrial conditions in the atmosphere-only model configuration closely resemble those of the fully coupled experiments. The temperature anomalies generally resemble previous Eemian simulations (i.e. the multi-model mean from Lunt et al., 2013), but exhibit larger warming in the Arctic and the North Atlantic region. The largest difference is found in the Nordic seas and the Labrador Sea, where our simulated annual mean warming is several degrees higher. Overall, the Arctic annual mean warming is less than 1 K higher than the multi-model mean. Figure 3 shows that entire Greenland warms in all seasons in the full Eemian experiment, iL+oL. Peak warming is generally found in the coastal regions, but also the central, high altitude Summit region warms more than 2 K in both

summer (June-July-August; JJA) and winter (December-January-February; DJF). During summer, strong warming patches are collocated with areas of albedo decrease due to loss of snow cover in coastal, low-elevation areas (not shown). The most prominent albedo change is found on the central west coast, south of Disko Bay, where the JJA mean surface albedo decreases by up to 0.4. Coastal areas in the northwest, central and northeastern regions exhibit albedo decreases of 0.1-0.2. The iP+oL simulation (ocean-only) shows similar but smaller magnitude JJA mean albedo changes, while iL+oP (insolation-only) has an almost unchanged JJA mean surface albedo. This difference illustrates that the increased shortwave absorption and subsequent larger sensible heat flux from the surface contributes to these local, near-coastal warming peaks in iL+oL and iP+oL.

The hybrid simulations, iP+oL and iL+oP, exhibit very different annual cycles of warming. The temperature changes in iL+oP follow the insolation anomalies with warming during summer and cooling in fall (September-October-November; SON) and winter. The summertime warming is widespread, while the winter cooling is limited to the southwestern part of Greenland.

The oceanic changes in iP+oL cause warming over entire Greenland, peaking in the colder seasons fall and winter. Warming due to sea ice loss peaks during winter, following increased turbulent heat flux from the ocean surface where the insulating sea ice layer is lost (in agreement with previous studies of sea ice loss; e.g. Pedersen et al., 2016a; Vihma, 2014). Previous studies show that the GrIS near-surface temperature is sensitive to sea ice changes in its vicinity (Pedersen et al., 2016a), and that ice loss in the Nordic Seas could have a larger impact than ice loss in the Labrador Sea due to an atmospheric circulation response (Merz et al., 2016). Additional SST increase from ocean circulation changes and increased summertime shortwave absorption (Pedersen et al., 2016b) expands the regions with positive turbulent heat flux anomalies beyond the areas of sea ice loss (not shown). The total impact of the oceanic changes thus counters the direct impact of the insolation during fall and winter, resulting in the all-year warming observed in iL+oL, which overall resembles the sum of the iP+oL and iL+oP (cf. Fig. 3, bottom row). The largest difference is found in JJA near the Disko Bay on the central west coast, where the iL+oL warming is stronger than the sum of the hybrid experiments. As previously described, this region exhibits an albedo decrease due to loss of snow cover. The combination of snow melt driven by oceanic warming and the positive insolation anomaly in iL+oL gives rise to a strengthened albedo feedback that causes the apparent non-linearity. The insolation anomaly alone (in iL+oP) only causes a modest loss of snow cover, and the impact of the surface albedo feedback is therefore limited.

Similar to the temperature, the snowfall over GrIS also exhibits varying sensitivity to the insolation and the oceanic changes. Figure 4 reveals several examples of contrasting snowfall changes on the east and western side of the ice divide, illustrating the barrier effect of the ice sheet (e.g. Ohmura and Reeh, 1991). This pattern suggests that inclusion of ice sheet topography changes could impact the precipitation patterns; as illustrated by Merz et al. (2014b). In iL+oL, the southern Greenland snowfall is increased throughout the year. The west coast appears drier during summer due to an increasing fraction of the precipitation falling as rain; the total precipitation is increased along the coast (not shown). The entire interior ice sheet receives more snow during summer, while the eastern (western) part have increased snowfall during fall (spring, March-April-May; MAM). The hybrid experiments reveal that the snowfall increase primarily is driven by the oceanic changes: iL+oL and iP+oL have high resemblance, especially winter and spring. The wintertime snowfall increase in the south-southeast is connected to a circulation anomaly with increased onshore winds on the southeast coast, which appears in both iL+oL and iP+oL (not shown). The increased onshore advection is expected to increase the precipitation due to orographic lifting (Ohmura and Reeh, 1991; Roe,

2005). During summer, iL+oP illustrates that the insolation contributes to the snowfall increase over the interior ice sheet. The fall pattern in iL+oL on the other hand indicates non-linear behavior, in that iL+oL does not resemble the sum of the two hybrid experiments: the increase on the eastern GrIS is only seen in iL+oL (as illustrated by difference in the bottom row of Fig. 4). Note, however, that Fig. 4 displays the relative change in snowfall, and the elevated northeastern region is very dry. The absolute snowfall anomaly (not shown) decreases rapidly towards the interior ice sheet, and the peak in the relative anomalies corresponds to a modest absolute increase (less than 0.2 mm/day). The non-linear behavior appears to be related to atmospheric circulation changes. The seasonal mean 10 m wind in iL+oL (not shown) has strengthened southward flow along the northeast coast compared to the control climate and the hybrid experiments, which could contribute to increased orographic precipitation in the region; despite being mainly along the coast. The modest absolute precipitation increase in this region could, however, be related to very few storm events, that would not be evident from the seasonal mean circulation anomalies.

## 3.1 Precipitation-weighted temperature

The precipitation changes in Fig. 4 suggest that the northwestern GrIS near the NEEM ice core location is affected by changed precipitation seasonality in all three simulations: the insolation in iL+oP causes increased summer snowfall and drier conditions in fall, the oceanic changes in iP+oL cause increased snowfall throughout the year, and the combination in iL+oL leads to increased snowfall during spring and summer. To assess how these changes might affect the ice core record, the precipitation-weighted annual mean temperature has been calculated following Eq. (1); Figure 5 compares the annual mean temperature change and the precipitation-weighted mean.

The precipitation-weighted mean temperature ($T_{pw}$) in iL+oL is relatively similar to the annual mean ($T_{ann}$). One exception is the northwestern GrIS, where $T_{pw}$ is higher suggesting a strong bias towards summer in the precipitation seasonality. The widespread, general precipitation increase driven by the changed oceanic conditions in iP+oL only causes minor differences between the annual and the precipitation-weighted means: The precipitation-weighting primarily affects the near-coastal regions. Conversely, iL+oP exhibits a large difference between $T_{ann}$ and $T_{pw}$. The combination of wetter summer conditions and drier conditions during winter and fall in western Greenland, increases $T_{pw}$ in a large region covering the central, west, and northwest GrIS. By comparing the hybrid simulations to the response in iL+oL, it appears that the insolation changes are responsible for the changed precipitation seasonality that causes the increased $T_{pw}$ in the northwestern GrIS. The difference between iL+oL and the sum of the hybrid experiments illustrate similar magnitude, but opposite differences for $T_{ann}$ and $T_{pw}$. The non-linear behavior here, in temperature as well as precipitation, is related to the varying response of the atmospheric circulation. The steep slopes of the ice sheet combined with katabatic winds and the anti-cyclonic circulation around the ice sheet margins generally limits the heat advection towards the interior ice sheet (Noël et al., 2014; Merz et al., 2016). Hence, potential precipitation and temperature changes on the interior ice sheet are largely dependent on changes in the circulation, which is not responding linearly to the combined insolation and oceanic forcings. The largest deviation is, however, found on the central west coast, where loss of snow cover and a strengthened albedo feedback explain the non-linearity (as described in relation to Fig. 3).

Due to the flow of the ice, the deposition site of the Eemian ice from the NEEM ice core is further upstream than the drilling location (approximately 76.4° N, 44.8° W, 205±20 km upstream; NEEM community members, 2013). To ensure fair comparison, we consider the simulated conditions over this point (dNEEM) when comparing model results and ice core records (note, however, that the upstream correction is within two grid cells). Table 2 presents the annual mean and precipitation-weighted temperatures for dNEEM, revealing a varying impact between the three simulations. The iL+oL estimate is largely unchanged by the precipitation-weighting, and is thus still relatively low compared to the ice core temperature reconstructions. As evident from Fig. 5, the precipitation-weighted temperature north-northwest of dNEEM exhibits a higher increase, but the strongest does not exceed 3 – 4 K. From the annual mean temperatures, the SST and sea ice changes (iP+oL) appear to completely dominate the temperature change at dNEEM. Taking the precipitation seasonality into account, however, reveals that the direct impact of the insolation has a comparable contribution to the warming signal recorded in the ice core.

As identified from Fig. 5, the three experiments indicate a non-linearity in the responses to the changed insolation and oceanic conditions at dNEEM. While iP+oL indicates that SST and sea ice changes results in a warming of 1.5 K ($\Delta T_{pw}$) at the NEEM location, the difference between iL+oL and iL+oP suggests that the oceanic changes only contribute with 0.8 K ($\Delta T_{pw}$) additional warming. Due to this non-linear relation between the experiments, we conclude that the warming impact of the oceanic changes is in the range 0.8 – 1.5 K. Similarly for $T_{ann}$, the oceanic changes contribute 1.9 – 2.5 K to the annual mean warming at dNEEM.

## 3.2   GrIS surface mass balance

The combined impact of the increasing temperature and the general precipitation increase is important for the response, and potential reduction, of the Greenland ice sheet. While the ice sheet is fixed in our experiments, offline calculations with the subsurface model presents an estimate of the combined effect of the simulated warming and precipitation changes over GrIS. Although the dynamical ice flow response is an important part of the total ice sheet mass balance (e.g. through calving), the surface mass balance (SMB) anomaly is a good indicator of potential ice sheet changes. The SMB is the sum of accumulation and ablation (i.e. run-off), which is here calculated in each grid cell based on 6-hourly output from the EC-Earth experiments (incoming shortwave and downward longwave radiation, latent and sensible heat fluxes, rain, snow and evaporation/sublimation). As EC-Earth does not have an explicit glacier mask, we have performed the calculations in all grid cells with a minimum snow depth of 10 cm across all the simulations. Consequently, our estimates likely do not capture the full ablation zone, where the snow cover would melt every year. Therefore, we do not consider the integrated SMB, but only the spatial pattern. Figure 6 shows the resulting SMB changes in the three simulations compared to iP+oP.

The SMB change in iL+oL reveals a general decrease along the coast extending further inland in the north combined with increased values along the southeast coast. The central, most elevated part of the ice sheet has a small SMB increase. The hybrid experiments reveal that most of the SMB reduction is caused directly by the insolation change (cf. iL+oP) with only a minor contribution from the oceanic conditions (mainly in the southwest, cf. iP+oL). The oceanic conditions, however, appear to drive the SMB increase in the southeast through a circulation change favoring increased orographic precipitation. The difference between iL+oL and the sum of iL+oP and iP+oL reveals non-linear responses along the western coast. In the

northwest, the combined forcing results in accelerated melt, while circulation related DJF snowfall increase in iL+oL (cf. Fig. 4) counters the summertime melt further south.

While no dynamic feedbacks of the ice sheet are included in these estimates, the reduced SMB in the north and northeast are consistent with ice sheet retreat in this region (in line with the reconstructions by Born and Nisancioglu, 2012; Quiquet et al., 2013; Stone et al., 2013). The increased SMB in the southeast further suggests that the ice sheet could persist in southern Greenland and remain connected to the main dome near the current summit, as the accumulation increase seems to overwhelm the impact of the increased temperature.

A rough estimate of the potential ice sheet height changes can be obtained by assuming that the simulated anomalies act over a relevant period; e.g. 1,000 years. In this view, the indicated anomalies on Fig. 6 can be read as "meters of water equivalent pr. 1,000 years". To account for the different density of the ice, the SMB values should be multiplied by $\frac{\rho_{\text{water}}}{\rho_{\text{ice}}} \approx 1.09$. Acting over 1,000 years, the extreme SMB values in Fig. 6 correspond to more than 550 m reduction near the southwest, northwest and northeastern coasts and more than 550 m growth near the southeastern coast.

In summary, our experiments indicate that the insolation change has a stronger impact on the GrIS SMB compared to the SST and sea ice changes. While the oceanic changes cause a larger increase of the annual mean temperature over Greenland (Fig. 5), the impact during summer and thus the contribution to increased melting is limited (cf. Fig. 3). The increased summer melt appears to be crucial for the GrIS SMB, as strong SMB reductions are evident in iL+oP, despite the fall and winter cooling. This importance of the Eemian insolation changes for the GrIS SMB, was previous illustrated by van de Berg et al. (2011) based on their regional climate model experiments separating the impacts of changed insolation and changed ambient climate. Despite the fact that the Arctic warming appears stronger in our experiments [comparing EC-Earth (Pedersen et al., 2016b) to the simulated Eemian warming in the ECHO-G model (Cubasch et al., 2006; Kaspar et al., 2007) used as boundary conditions in van de Berg et al. (2011)], the direct impact of the insolation is still the dominant contribution to the GrIS SMB changes. Our results thus similarly illustrate that the relation between warming and GrIS melting during the Eemian is likely not suitable for estimating the ice sheet response to future, greenhouse gas-driven warming. The combined effect of the more seasonally uniform warming and the general snowfall increase driven by the oceanic changes (iP+oL) results in a less pronounced SMB response. In southeastern Greenland, however, our experiments indicate that the oceanic changes dominate the SMB response through the increased snowfall and accumulation (Fig. 6).

## 4   Conclusions

Our experiments suggest that Greenland experienced higher temperatures and increased snowfall throughout the year during the Eemian. The hybrid simulations, which compare the direct impact of the insolation change to the indirect impact of changed sea surface conditions, illustrate that the largest contribution to both the warming and snowfall increase is due to oceanic changes. The ocean-only experiment exhibits increased temperatures and snowfall throughout the year, and provides a memory effect that prolongs the impact of the summertime insolation increase. The warming is widespread and even reaches the

interior, elevated part of the ice sheet (including the NEEM ice core site). The direct impact of the insolation favors increased temperature and snowfall during summer, but colder and drier conditions during fall and winter.

Analysis of the simulations from an ice core perspective changes the apparent relative importance of insolation and sea surface changes. At the NEEM deposition site, the insolation changes favor changes in the precipitation seasonality that increase the summer weight in the precipitation-weighted temperature. Consequently, the isolated impacts of insolation and the associated oceanic changes on the precipitation-weighted temperature are comparable, despite the fact that the oceanic changes cause about 2 K higher annual mean warming. With the precipitation seasonality taken into account, our simulations, in line with previous model studies (Braconnot et al., 2012; Lunt et al., 2013; Masson-Delmotte et al., 2013; Otto-Bliesner et al., 2013), underestimate the warming compared to the NEEM ice core reconstructions.

The model–data discrepancy can in part be explained by the experiment design and the model performance. Incorporation of ice sheet elevation changes would increase the near-surface temperature following the atmospheric lapse rate, and local circulation might furthermore be affected by altered ice sheet topography. Merz et al. (2014a) estimate that an increased slope of the ice sheet could contribute up to 3.1 K annual mean warming locally at the NEEM location. Additionally, changed precipitation patterns could affect the precipitation-weighted temperature reflected in the ice core record (Merz et al., 2014b).

Sea ice changes can affect both warming, circulation, and precipitation near GrIS depending on both the magnitude and the location of sea ice loss (Merz et al., 2016; Pedersen et al., 2016a). EC-Earth simulates an extensive sea ice cover under present and pre-industrial conditions, suggesting that the Eemian sea ice cover is similarly overestimated. Further sea ice reduction, especially in the vicinity of Greenland, could further increase GrIS warming (Pedersen et al., 2016a). Based on precipitation-weighted temperature estimates, our hybrid simulations indicate that the combined effect of sea ice loss and SST increase is responsible for 0.8–1.5 K warming recorded at the NEEM deposition site (annual mean temperatures indicate a warming impact of 1.9–2.5 K). Merz et al. (2016) illustrate how further sea ice reduction could accelerate Greenland warming, and estimate that uncertainty in the sea ice cover can account for 1.6 K annual mean warming at the NEEM site.

The combined impact of the simulated warming and snowfall increase could favor substantial ice sheet changes. SMB calculations revealed that while the oceanic changes favor increased accumulation over the southeastern GrIS, the changed insolation causes increased melting along the coastal parts of the ice sheet. The hybrid experiments indicate that the insolation is the dominant factor behind the expected reduction of the GrIS. This reiterates the finding of van de Berg et al. (2011), that direct use of the relation between temperature and mass loss in the Eemian is likely to overestimate future greenhouse gas-driven melting. The SMB changes are consistent with previous ice sheet reconstructions (e.g. Born and Nisancioglu, 2012; Quiquet et al., 2013; Stone et al., 2013) suggesting ice sheet retreat in the southwest and northern coastal GrIS.

*Acknowledgements.* The authors thank Valérie Masson-Delmotte for valuable discussions. Acknowledgement is made for the use of ECMWF's computing and archive facilities in this research. The research leading to these results has received funding from the European Research Council under the European Union's Seventh Framework Programme (FP7/2007–2013)/ERC Grant Agreement 610055 as part of the ice2ice project. The authors acknowledge the support of the Danish National Research Foundation through the Centre for Ice and

Climate at the Niels Bohr Institute. We thank the editor and two anonymous reviewers for constructive suggestions that helped improve the manuscript.

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

**Tables**

**Table 1.** Boundary conditions for the experiments. In the experiment names, the letter following "i" indicates the insolation conditions, while the letter following "o" indicates the oceanic conditions: "P" is PI and "L" is Eemian (Last Interglacial).

| Experiment | Insolation and GHGs | SSTs and sea ice |
|---|---|---|
| iP+oP | Pre-industrial | Pre-industrial |
| iL+oL | Eemian | Eemian |
| iL+oP | Eemian | Pre-industrial |
| iP+oL | Pre-industrial | Eemian |

**Table 2.** Annual mean ($T_{\mathrm{ann}}$) and precipitation-weighted ($T_{\mathrm{pw}}$) temperature change relative to iP+oP and associated standard deviations ($\sigma_{\mathrm{ann}}$, $\sigma_{\mathrm{pw}}$) for dNEEM. *Not statistically significant at the 95 % confidence level.

| Experiment | $\Delta T_{\mathrm{ann}}$ | $\sigma_{\mathrm{ann}}$ | $\Delta T_{\mathrm{pw}}$ | $\sigma_{\mathrm{pw}}$ |
|---|---|---|---|---|
| iL+oL | **2.3 K** | 1.5 K | **2.4 K** | 3.1 K |
| iL+oP | **-0.2 K**\* | 1.3 K | **1.6 K** | 2.7 K |
| iP+oL | **1.9 K** | 1.4 K | **1.5 K** | 2.6 K |

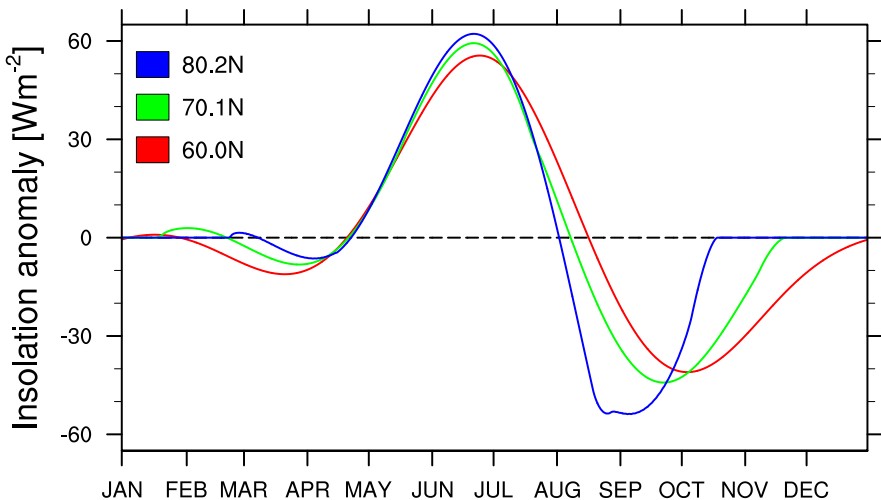

**Figure 1.** Insolation anomalies [Wm$^{-2}$] in Eemian relative to PI. The selected latitudes represent southern (60° N, red), middle (70° N, green), and northern (80° N, blue) Greenland. Tick marks indicate the beginning of each month.

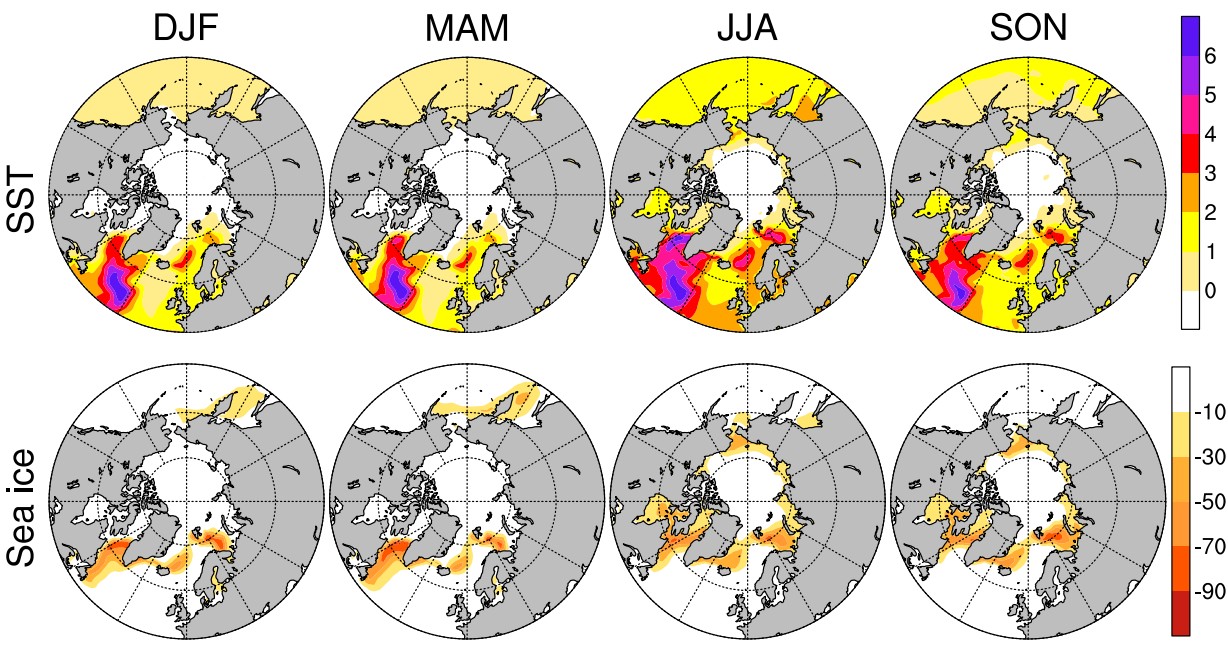

**Figure 2.** Seasonal mean (top) SST [K] and (bottom) sea ice concentration anomalies [%] in Eemian relative to PI boundary conditions.

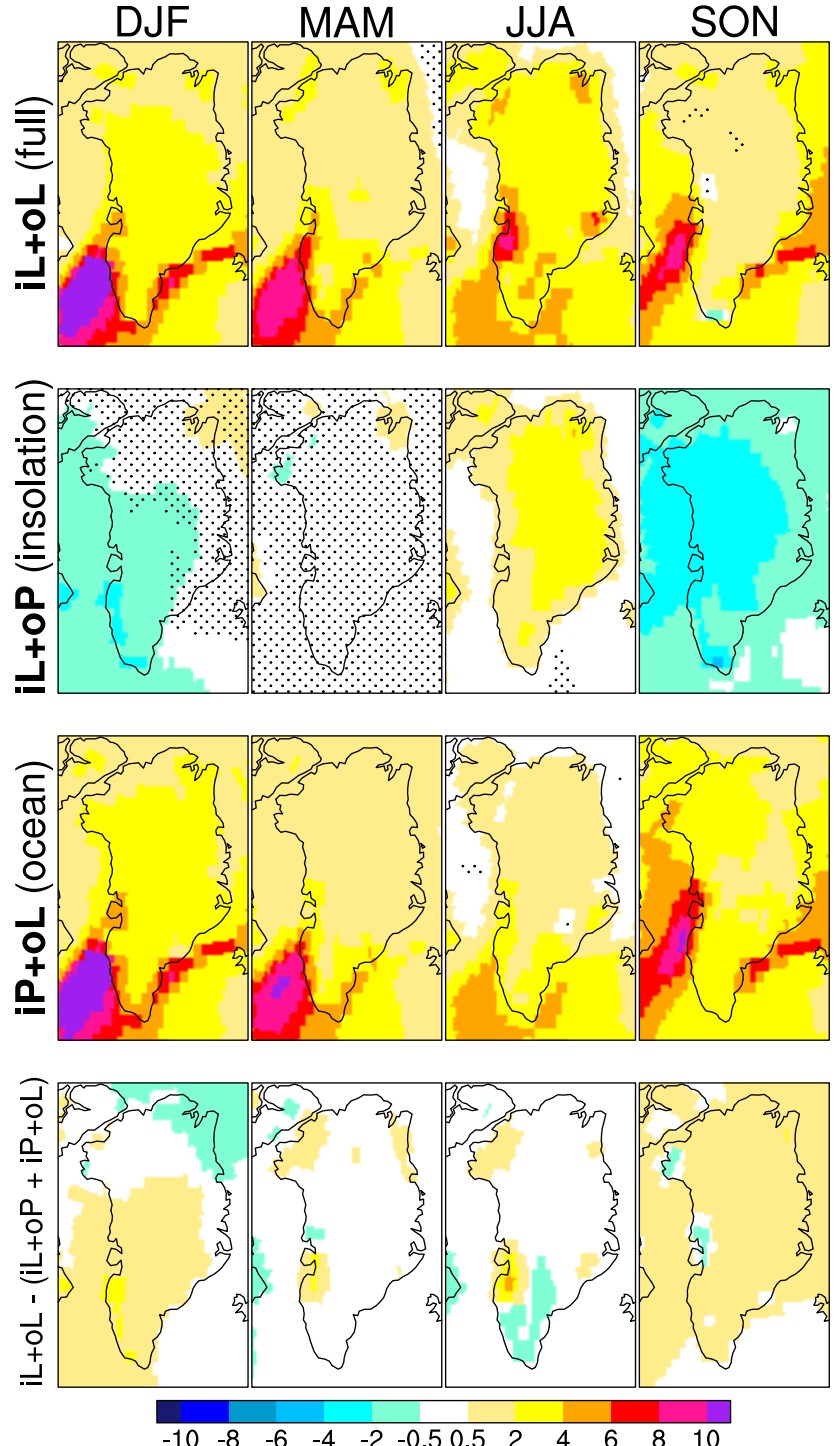

**Figure 3.** Seasonal mean near-surface temperature anomalies [K] compared to iP+oP in the three experiments: **iL+oL** (top row), **iL+oP** (second row), **iP+oL** (third row), and the difference **iL+oL − (iL+oP + iP+oL)** (bottom row). Black dotted shading marks anomalies that are not statistically significant at the 95 % level.

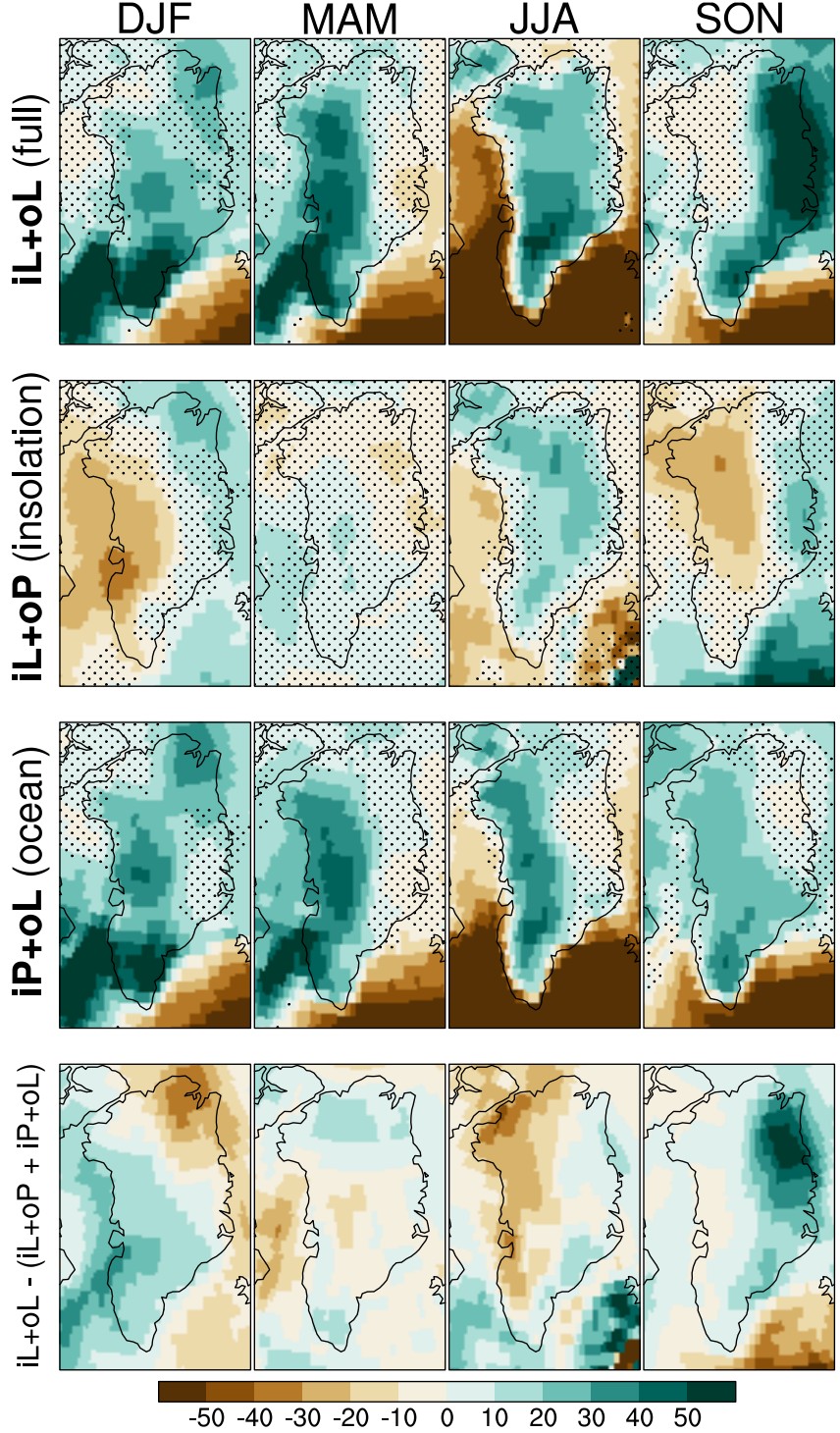

**Figure 4.** Seasonal mean relative snowfall anomalies [%] from iP+oP: **iL+oL** (top row), **iL+oP** (second row), **iP+oL** (third row), and the difference **iL+oL − (iL+oP + iP+oL)** (bottom row). Black dotted shading marks anomalies that are not statistically significant at the 95 % level.

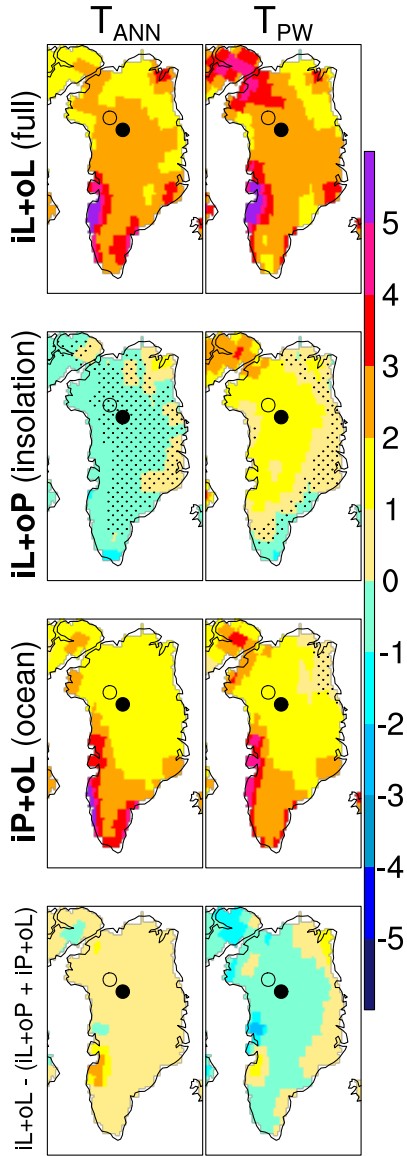

**Figure 5.** Near-surface air temperature anomalies [K] relative to iP+oP in **iL+oL** (top row), **iL+oP** (second row), **iP+oL** (third row), and the difference **iL+oL − (iL+oP + iP+oL)** (bottom row): Annual mean (left column) and precipitation-weighted mean (right column). Black dotted shading marks anomalies that are not statistically significant at the 95 % level. dNEEM (NEEM) location is marked with the filled (hollow) black circle. Note the changed color bar compared to the seasonal means.

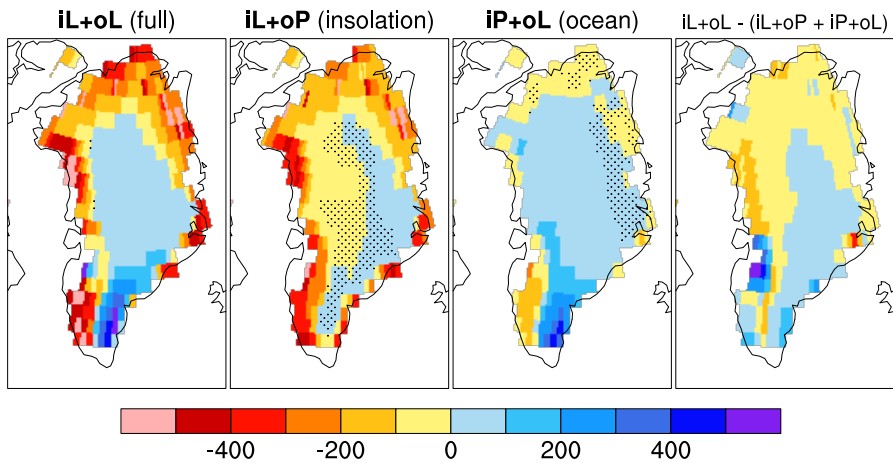

**Figure 6.** Surface mass balance anomalies [mm water-equivalent] compared to iP+oP. From left to right: **iL+oL**, **iL+oP**, **iP+oL**, and the difference **iL+oL − (iL+oP + iP+oL)**. Black dotted shading marks anomalies that are not significant at the 95 % confidence level.