# Peer review of "Greenland during the last interglacial: the relative importance of insolation and oceanic changes"

_Climate of the Past, 2016_

## Referee Comment (RC1) · Anonymous Referee #1 · 16 May 2016

The manuscript presents an interesting analyses of a long standing question 'how can we explain the large amount of reconstructed LIG Greenland warming relative to climate model simulations?'. By presenting multiple simulations with different forcing and boundary conditions a clean investigation of the role of different forcings can be made. Although I think the manuscript should be clearer at some points, and more in depth at others, I deem it suited for publications with minor revisions.

Main comments: In its current form it is not clear to me why calculations of SMB are included. This part of the manuscript should be better introduced and embedded. Moreover, is it possible to use the SMB calculations to provide a rough estimate of the amount of surface elevation change that would result during the LIG? This would be

a great addition to the presented work and clearly show the added value of including SMB calculations.

In the presented study, a fixed seasonal calendar is used. Although common practice in palaeoclimate modelling, it seems to me that in this study the biases that this assumption introduces might be of importance. Several publications have shown that the impact of using a fixed seasonal calendar rather than a fixed angular calendar has largest impact during LIG NH autumn. The results presented in this study have a strong focus on seasonal changes, thus it should be explained what the possible impact of using a fixed seasonal calendar could be.

The presented study uses oceanic boundary conditions from a previous coupled simulation. Although it is outside the scope of this study to discuss in detail what caused the imposed changes in ocean surface temperatures and sea-ice cover, a short description of the relevant changes, causes and uncertainties is important to interpret the presented results. More specifically I'm thinking about the changes in the AMOC strength, sea-ice changes and changes in deep convection in the North Atlantic region. What drives those? Are there indications from palaeo climate reconstructions that such changes are realistic and how does that impact the presented results? The pre-industrial sea-ice cover in EC-Earth seems rather extensive, does that impact the LIG results?

It is described that the simulated changes at NEEM are not in accordance with reconstructions. Related to that, how does the EC-Earth simulation compare to previous model results (for instance Lunt et al, 2012, Landais et al. 2016), could these differences partly explain the mismatch between EC-Earth LIG NEEM temperatures and reconstructions?

Minor comments: In the manuscript the terms direct and indirect are often used, are they in this context the same as forced response and feedbacks? Please clarify and consider updating in the manuscript.

Page 1 lines 12-14: The link of this section to the title of the manuscript is not clear. Please clarify.

Page 2 line 1: Consider including Landais et al. 2016 at this point since that study also compares model results specifically for Greenland.

Page 2 line 20: Is the suggested elevation change really limited. The given uncertainty could also indicate that in fact the changes where on the order of 300m, implying a temperature effect on the order of 2 degrees.

Page 2 line 24: 'Local conditions' is a little vague, what are you referring to, please clarify.

Page 3 lines 28-29: 'and consequently..' this is perhaps too obvious. Consider removing.

Page 4 line 20: 'Ice sheets….' This has been mentioned before, consider removing.

Page 5 lines 9-11: Are these areas of snow cover changes over the ice sheet or over vegetated land, and related, how large is the albedo change that likely caused these patches of strong warming?

Page 6 lines 3-4: What kind of non-linear processes are causing this behaviour? Please clarify.

Perhaps not so much a comment as a question: it appears that precipitation and snow have an almost one-to-one relationship, higher temperatures result in more snow fall, is that correct? Is that simply because warm air can hold more moisture, or is there also an effect from changes in sea-ice cover and moisture availability?

Table 2: The sum of the iL+oP and iP+oL experiments is quite different from the iL+oL result. Is this because of non-linear effects and if so, what are they? Internal variability?

Technical comments:

Throughout the text double brackets are found in references. Perhaps a latex issue, but consider using only single brackets.

The word 'thus' is used a little often, consider replacing with synonyms or restructuring some sentences.

Page 3 line9: remove 'the' before GrIS.

Page 3 line 12: remove 'the' twice in this sentence.

Page 5 line 7: Remove 'the' in front of 'peak'.

Page 5 line 7: perhaps 'but also the central, high...'?

Page 5 line 13: Confusing sentence, please rewrite.

Page 5 line 29: Please rewrite because from this it could be understood that topography changes are driving precipitation changes, while no such topography changes are applied in this study.

Page 6 line 4: should this be eastern GrIS?

Page 8 line 1: 'appear', are they or are they not in your experiments?

Figure 1: the curve for 80.2N has a spurious kink between August and September, is this real or somehow an artefact of the calculations or plotting?

Figure 2: the direction labels are unreadable. Perhaps the figures can be made larger in general because the sea-ice extents are also difficult to see.

Figure 3: why are only changes larger than 0.5K shown, it appears to me that they are shown in white.

Figure 3 and 4: Perhaps in both figures the sum of iL+oP and iP+oL can also be shown to compare to iL+oL in order to highlight the non-linearities? Or the difference between the sum and iL+oL?

Figure 4 and 5: Why are the ocean areas masked out?

Figure 5: It seems the annual mean temperatures are even more non-linear when considering the individual and combined forcings. Please discuss.

Figure 6: remove 'iP+oP' from titles in line with the other figures.

---

## Referee Comment (RC2) · Anonymous Referee #2 · 24 Jun 2016

This manuscript evaluates the role of insulation and sea surface temperature changes on the Greenland temperature during the Eemian. This work is interesting and valuable as it could offer insights into (i) the drivers of sea level changes during that period, (ii) the drivers of climate change and (iii) the reasons for the discrepancy between modelled and reconstructed Greenland temperature. The work carried out is sound and well described (apart from a few minor clarifications that need to be made), but the implications of the results are not sufficiently well presented and some of the analysis needs to go a bit further. This paper could have a lot more impact with a little bit of adjustment to the manuscript and a little bit more analysis of the result. I therefore suggest the manuscript to be accepted after some corrections and clarification. These

would be a bit more than minor revisions, but i don't anticipate they would require too much work.

Overall the manuscript does a good job at describing the changes associated with SST and insulation forcings, but does discuss the reasons of these changes. In particular, I would like to see some explanation of the role of insulation on precipitation seasonality.

There is some mention of 'non-linearity' effect, but this is very much glanced over. It needs more description of what that means, how strong the non-linearity is and what causes it.

In the discussion and introduction, clarify that part of the SST changes are caused by insulation and that this study focuses on the direct effect of insulation vs ocean temperature changes. Also, there should be a discussion of how well the model simulates modern Greenland temperature and how that would impact interpretation of the results. For example, some GCMs have difficulties simulating Arctic cloud processes. Could that affect the sensitivity of the model to changes in insulation/SSTs ?

The conclusions of the manuscript are a bit underwhelming. The start of the manuscript suggests that this study could shed light on the reasons for model-data discrepancy regarding Greenland Eemian temperature. The paper concludes that changes in ice sheet topography are to be blamed, but that is precisely a factor that the paper was not including. Is there nothing to be learned about the model's sensitivity to insulation and SST changes ?

Finally the mass balance calculations are really interesting and valuable, but the results are a bit lost in the manuscript which is a real shame.

Other minor comments: Line 20: "While the ice core air content only suggests limited elevation changes at the NEEM site (45±350 m higher than present ice sheet elevation), the NEEM ice core temperature reconstruction has been corrected using the surface elevation change estimate from the ice core air content (NEEM commu-

nity members, 2013)." A lot of repetition in this sentence which I find a bit difficult to understand, so I suggest modifying it.

Section 2.3 page 2, line 20. Reference for the SST and sea ice boundary conditions. Is this from Pedersen et al. (2016b)?

Section 2.3 line 27: clarify, what the impact of insolation on SST changes is based on ? is this again from Pedersen et al. (2016b) ?

Page 5, line 29: "The simulated responses reveal that the ice sheet topography is important for the precipitation changes: Figure 4 30 reveals several examples of contrasting snowfall changes on the east and western side of the ice divide." I understand what is meant here, but I would suggest clarifying this statement as the readers may confuse (i) the control that topography has on the pattern of climate change observed, with (ii) the effect of topographical changes not included here.

Figure 6 add label for "effect of SST" "effect of insolation" above the subplots to help the reader understand the results.

Page 6, line 31. This paragraph needs more discussion. The second sentence is not enough to justify the non-linearity. I suggest formalising slightly more the factor decomposition to calculate the interaction between ocean and insolation forcings (see Stein and Alpert) or at least state that adding the two effects does not give the full temperature change. Also, add a discussion of the reasons for this. Why is this nonlinearity different for precipitation-weighted and absolute temperature difference? Can you explain the processes that lead to the non-linearity ?

Stein, U., Alpert, P., 1993. Factor Separation in Numerical Simulations. Journal of the Atmospheric Sciences 50, 2107–2115.

---

## Author Comment (AC1) · 22 Aug 2016

**Authors' response to reviewer comments on "Greenland warming during the last interglacial: the relative importance of insolation and oceanic changes" by Rasmus A. Pedersen et al.**

We would like to thank the reviewers for the constructive comments. The authors' response to each comment is inserted below in *blue italics.*

Page and line numbers below mark the locations in the attached marked-up manuscript.

**Anonymous Referee #1**

[Figure]

The manuscript presents an interesting analyses of a long standing question 'how can we explain the large amount of reconstructed LIG Greenland warming relative to climate model simulations?'. By presenting multiple simulations with different forcing and boundary conditions a clean investigation of the role of different forcings can be made. Although I think the manuscript should be clearer at some points, and more in depth at others, I deem it suited for publications with minor revisions.

**Main comments**: In its current form it is not clear to me why calculations of SMB are included. This part of the manuscript should be better introduced and embedded. Moreover, is it possible to use the SMB calculations to provide a rough estimate of the amount of surface elevation change that would result during the LIG? This would be a great addition to the presented work and clearly show the added value of including SMB calculations.

*The introduction has been expanded with a paragraph on SMB.*

*Page 3, Line 24*

*"In the assessment of the Greenland climate, we also aim to investigate how the simulated Eemian climate could impact the ice sheet. The ice sheet response is a combined result of dynamics (ice flow) and surface mass balance changes (melt and accumulation). Here, we employ a detailed surface scheme to assess the surface mass balance. Again, the assessment of the relative importance of the insolation and sea surface warming will indicate whether Eemian ice sheet reconstructions are useful analogues for future ice sheet changes."*

*A realistic estimate of the height change would require a dynamical contribution in addition to the SMB. Nevertheless, we have added a rough estimate translating the values in Fig. 6 into height changes by assuming that the simulated changes act over 1,000 years.*

*Page 9, Line 30*

[Figure]

*"A rough estimate of the potential ice sheet height changes can be obtained by assuming that the simulated anomalies act over 1,000 years. In this view, the indicated anomalies on Fig. 6 can be read as "meters of water equivalent pr. 1,000 years". To account for the different density of the ice, the SMB values should be multiplied by $\frac{\rho_{water}}{\rho_{ice}} \approx 1.09$. Acting over 1,000 years, the extreme SMB values in Fig. 6 correspond to more than -550 m reduction near the southwest, northwest and northeastern coasts and more than 550 m growth near the southeastern coast."*

In the presented study, a fixed seasonal calendar is used. Although common practice in palaeoclimate modelling, it seems to me that in this study the biases that this assumption introduces might be of importance. Several publications have shown that the impact of using a fixed seasonal calendar rather than a fixed angular calendar has largest impact during LIG NH autumn. The results presented in this study have a strong focus on seasonal changes, thus it should be explained what the possible impact of using a fixed seasonal calendar could be.

*We have added a comment on the calendar and season definitions.*

*Page 5, Line 17*

*"The presented results are all following a fixed seasonal calendar. Compared to an alternative angular calendar based on astronomical positions, this changes the seasonal variation of the insolation. As the calendar is defined with a fixed vernal equinox, the largest difference between the two calendar definitions is found during northern hemisphere autumn. Joussaume and Braconnot (1997) illustrate that the negative anomaly at high northern latitudes (cf. Fig. 1) does not appear with angular season definitions. Note, however, that annual mean anomalies are not affected by the choice of calendar.*
*"*

The presented study uses oceanic boundary conditions from a previous coupled simulation. Although it is outside the scope of this study to discuss in detail what caused the imposed changes in ocean surface temperatures and sea-ice cover, a short description of the relevant changes, causes and uncertainties is important to interpret the presented results. More specifically I'm thinking about the changes in the AMOC strength, sea-ice changes and changes in deep convection in the North Atlantic region. What drives those? Are there indications from palaeo climate reconstructions that such changes are realistic and how does that impact the presented results? The pre-industrial sea-ice cover in EC-Earth seems rather extensive, does that impact the LIG results?

*We have added a few headlines describing the oceanic boundary conditions derived from the previous coupled simulations. The changes are covered in detail in the companion paper Pedersen et al. (2016b), which has now been published. As suggested, the pre-industrial simulation has an extensive sea ice cover and no active deep convection in the Labrador Sea. The activation of the Labrador Sea convection contributes to an increased AMOC strength and the general North Atlantic warming seen in Figure 2.*

*Page 5, Line 23*

*"In the coupled simulations from Pedersen et al. (2016b), the induced insolation forcing leads to sea ice retreat and increasing SSTs across high northern latitudes. Figure 2 depicts sea ice concentration and SST anomalies in the coupled simulations from Pedersen et al. (2016b), indicating the differences between the sea surface boundary conditions employed here. The sea ice reduction is primarily manifested as a northward retreat of the ice edge; the sea ice concentration in the central Arctic is largely unchanged. The strongest warming is found in the North Atlantic following the northward retreat of the sea ice edge and a strengthening of the Atlantic meridional overturning circulation (AMOC). The AMOC increase is related to a regional increase of the surface salinity and increased wintertime convection, which is in part related to biases in the pre-industrial control climate (see detailed description in Pedersen et al., 2016b)."*

It is described that the simulated changes at NEEM are not in accordance with reconstructures. Related to that, how does the EC-Earth simulation compare to previous model results (for instance Lunt et al, 2012, Landais et al. 2016), could these differences partly explain the mismatch between EC-Earth LIG NEEM temperatures and reconstructions?

*We have added a comment comparing the simulation to previous work.*

*Page 6, Line 5*

*"As described by Pedersen et al. (2016b), the anomalies between Eemian and pre-industrial conditions in the atmosphere-only model configuration closely resemble those of the fully coupled experiments. The temperature anomalies generally resemble previous Eemian simulations (i.e. the multi-model mean from Lunt et al. 2013), but exhibit larger warming in the Arctic and the North Atlantic region."*

**Minor comments**:

In the manuscript the terms direct and indirect are often used, are they in this context the same as forced response and feedbacks? Please clarify and consider updating in the manuscript.

*We do not think that "forced response" and "feedback" are appropriate terms for the separation in our experiments. As an example, land points are free to respond to the insolation forcing in the "insolation-only" experiment (iL+oP). Hence, this response includes feedbacks as well. The description in the introduction has been updated to clarify the separation.*

*Page 3, Line 14*

*"Using a series of general circulation model (GCM) experiments, we assess the Greenland climate during the Eemian. We investigate how the simulated changes could affect the GrIS surface mass balance and the ice core record, and compare the reconstructed and simulated temperatures. While the insolation change is the only forcing in our experiments, we further compare the direct impact of the insolation change*

*and the indirect effect of retreating sea ice and increasing sea surface temperatures (SSTs). The direct and indirect impacts are separated using two hybrid experiments: one forced by Eemian insolation and fixed pre-industrial sea surface conditions (direct impact, "insolation-only") and one with pre-industrial insolation and Eemian sea surface conditions (indirect impact, "ocean-only"). The temperature change during the Eemian resembles that of future climate scenarios (e.g. Clark and Huybers, 2009), and our comparison could reveal whether the Eemian is an appropriate analogue for future climate change in Greenland; i.e. whether insolation or the ambient oceanic warming dominates the total response."*

- Page 1 lines 12-14: The link of this section to the title of the manuscript is not clear. Please clarify.

*The title has been modified to "Greenland during the last interglacial: the relative importance of insolation and oceanic changes". The lines have further been re-written.*

*Page 1, Line 12*

*"Surface mass balance calculations with an energy balance model further indicate that the combination of temperature and precipitation anomalies leads to potential mass loss in the north and southwestern parts of the ice sheet. The oceanic conditions favor increased accumulation in the southeast, while the insolation appears to be the dominant cause of the expected ice sheet reduction. Consequently, the Eemian is not a suitable analogue for future ice sheet changes."*

- Page 2 line 1: Consider including Landais et al. 2016 at this point since that study also compares model results specifically for Greenland.

*We have added Landais et al. 2016 to the listed references.*

- Page 2 line 20: Is the suggested elevation change really limited. The given uncertainty could also indicate that in fact the changes where on the order of 300m, implying a temperature effect on the order of 2 degrees.

*The sentence has been re-written.*

*Page 2, Line 24*

*"The NEEM ice core temperature reconstruction has been corrected for this effect using the ice core air content which suggests an elevation increase of 45±350 m relative to the present ice sheet elevation (NEEM community members, 2013)."*

- Page 2 line 24: 'Local conditions' is a little vague, what are you referring to, please clarify.

*'Local conditions' has been replaced with 'the surface energy balance on Greenland'.*

*Page 2, Line 27*

*"Besides the direct impact of the elevation change, changes in the GrIS topography could also impact the large scale circulation (Hakuba et al., 2012; Lunt et al., 2004; Petersen et al., 2004), as well as surface energy balance on Greenland (Merz et al., 2014a) and the ice core record through changed precipitation patterns (Merz et al., 2014b). "*

- Page 3 lines 28-29: 'and consequently..' this is perhaps too obvious. Consider removing.

*For clarity, we would like to stress that the feedbacks are disabled. We have previously encountered questions on this matter. The sentence has, however, been rewritten.*

*Page 4, Line 19*

*"Due to the diverse ice sheet reconstructions, we have kept the ice sheets fixed at present day extents in all of our simulations. Vegetation is similarly kept at present day values. Combined with the fixed SST and sea ice conditions, this means that our experiments do not include any additional feedbacks from the ocean, vegetation or ice sheet geometry (as discussed in Pedersen et al., 2016b)."*

- Page 4 line 20: 'Ice sheets: : :.' This has been mentioned before, consider removing.

*This has been removed.*

- Page 5 lines 9-11: Are these areas of snow cover changes over the ice sheet or over vegetated land, and related, how large is the albedo change that likely caused these patches of strong warming?

*We have elaborated the albedo discussion and added values for the surface albedo changes.*

*Page 6, Line 10*

*"During summer, strong warming patches are collocated with areas of albedo decrease due to loss of snow cover in coastal, low-elevation areas (not shown). The most prominent albedo change is found on the central west coast, south of Disko Bay, where the JJA mean surface albedo decreases by up to 0.4. Coastal areas in the northwest, central and northeastern regions exhibit albedo decreases of 0.1-0.2. The iP+oL simulation (ocean-only) shows similar but smaller magnitude JJA mean albedo changes, while iL+oP (insolation-only) has an almost unchanged JJA mean surface albedo. This difference illustrates that the increased shortwave absorption and subsequent larger sensible heat flux from the surface contributes to these local, near-coastal warming peaks in iL+oL and iP+oL."*

- Page 6 lines 3-4: What kind of non-linear processes are causing this behaviour? Please clarify.

*We have added new figures illustrating the non-linear response, i.e. the difference iL+oL – (iL+oP + iP+oL). Additionally, this particular discussion on the snowfall has been expanded.*

*Page 7, Line 21*

*"The fall pattern in iL+oL on the other hand indicates non-linear behavior, in that iL+oL*

*does not resemble the sum of the two hybrid experiments: the increase on the east-
ern GrIS is only seen in iL+oL (as illustrated by difference in the bottom row of Fig.
4). Note, however, that Fig. 4 displays the relative change in snowfall, and the el-
evated northeastern region is very dry. The absolute snowfall anomaly (not shown)
decreases rapidly towards the interior ice sheet, and the peak in the relative anomalies
corresponds to a modest absolute increase (less than 0.2 mm/day). The non-linear be-
havior appears to be related to atmospheric circulation changes. The seasonal mean
10 m wind in iL+oL (not shown) has strengthened southward flow along the northeast
coast compared to the control climate and the hybrid experiments, which could con-
tribute to increased orographic precipitation in the region; despite being mainly along
to coast. The modest absolute precipitation increase in this region could, however, be
related to very few storm events, that would not be evident from the seasonal mean
circulation anomalies. "*

- Perhaps not so much a comment as a question: it appears that precipitation and snow
have an almost one-to-one relationship, higher temperatures result in more snow fall,
is that correct? Is that simply because warm air can hold more moisture, or is there
also an effect from changes in sea-ice cover and moisture availability?

*We assume that "precipitation and snow" is a typo, and should be "temperature and
snow"? We expect that the increased moisture from local sources is a minor contri-
bution – large-scale circulation and moisture transport changes appear to be a more
dominant factor.*

- Table 2: The sum of the iL+oP and iP+oL experiments is quite different from the iL+oL
result. Is this because of non-linear effects and if so, what are they? Internal variability?

*This discussion has been expanded in relation to the new figures.*

*Page 8, Line 11*

*"The difference between iL+oL and the sum of the hybrid experiments illustrate similar*

*magnitude, but opposite differences for $T_{ann}$ and $T_{pw}$. The non-linear behavior here, in temperature as well as precipitation, is related to the varying response of the atmospheric circulation. The steep slopes of the ice sheet combined with katabatic winds and the anti-cyclonic circulation around the ice sheet margins generally limits the heat advection towards the interior ice sheet (Noël et al., 2014; Merz et al., 2016). Hence, potential precipitation and temperature changes on the interior ice sheet are largely dependent on changes in the circulation, which is not responding linearly to the combined insolation and oceanic forcings. The largest deviation is, however, found on the central west coast, where loss of snow cover and a strengthened albedo feedback explain the non-linearity (as described in relation to Fig. 3). "*

*Page 8, Line 31*

*"As identified from Fig. 5, the three experiments indicate a non-linearity in the responses to the changed insolation and oceanic conditions at dNEEM. While iP+oL indicates that SST and sea ice changes results in a warming of 1.5 K ($\Delta T_{pw}$) at the NEEM location, the difference between iL+oL and iL+oP suggests that the oceanic changes only contribute with 0.8 K ($\Delta T_{pw}$) additional warming. Due to this non-linear relation between the experiments, we conclude that the warming impact of the oceanic changes is in the range 0.8 − 1.5 K. Similarly for $T_{ann}$, the oceanic changes contribute 1.9 − 2.5 K to the annual mean warming at dNEEM."*

**Technical comments**:

Throughout the text double brackets are found in references. Perhaps a latex issue, but consider using only single brackets.

*The references have been corrected.*

The word 'thus' is used a little often, consider replacing with synonyms or restructuring some sentences.

*The use of 'thus' has been reduced.*

- Page 3 line 9: remove 'the' before GrIS.

*This has been removed.*

- Page 3 line 12: remove 'the' twice in this sentence.

*This has been removed.*

- Page 5 line 7: Remove 'the' in front of 'peak'.

*This has been removed.*

- Page 5 line 7: perhaps 'but also the central, high: : :'?

*This has been changed.*

- Page 5 line 13: Confusing sentence, please rewrite.

*The sentence has been re-written.*

*Page 6, Line 19*

*"The temperature changes in iL+oP follow the insolation anomalies with warming during summer and cooling in fall (September-October-November; SON) and winter. The summertime warming is widespread, while the winter cooling is limited to the southwestern part of Greenland."*

- Page 5 line 29: Please rewrite because from this it could be understood that topography changes are driving precipitation changes, while no such topography changes are applied in this study.

*The sentence has been re-written.*

*Page 7, Line 9*

*"Figure 4 reveals several examples of contrasting snowfall changes on the east and western side of the ice divide, illustrating the barrier effect of the ice sheet (e.g. Ohmura and Reeh, 1991). This pattern suggests that inclusion of ice sheet topography changes*

*could impact the precipitation patterns; as illustrated by Merz et al. (2014b)."*

- Page 6 line 4: should this be eastern GrIS?

*Yes; this has been corrected.*

- Page 8 line 1: 'appear', are they or are they not in your experiments?

*This has been rewritten.*

*Page 10, Line 13*

*"In southeastern Greenland, however, our experiments indicate that the oceanic changes dominate the SMB response through the increased snowfall and accumulation (Fig. 6). "*

- Figure 1: the curve for 80.2N has a spurious kink between August and September, is this real or somehow an artefact of the calculations or plotting?

*We have re-assessed the calculations and the absolute insolation curves, which exhibit no kinks. It appears to be a real feature related to varying rates of change in the high latitude insolation near the fall equinox.*

- Figure 2: the direction labels are unreadable. Perhaps the figures can be made larger in general because the sea-ice extents are also difficult to see.

*Figure 2 has been revised: SST and sea ice anomalies are now presented in separate panels. Directional labels have been removed to allow increased plot size.*

- Figure 3: why are only changes larger than 0.5K shown, it appears to me that they are shown in white.

*The caption has been revised. The comment was added to highlight the non-linear behavior of the color bar near 0.*

- Figure 3 and 4: Perhaps in both figures the sum of iL+oP and iP+oL can also be shown to compare to iL+oL in order to highlight the non-linearities? Or the difference

between the sum and iL+oL?

*The difference iL+oL – (iL+oP + iP+oL) has been added to all relevant figures (Figures 3-6).*

- Figure 4 and 5: Why are the ocean areas masked out?

*Ocean points have been added on Figure 4.*

*Figure 5 presents the precipitation-weighted temperature mean which is only meaningful on the ice sheet. Hence, ocean points are masked out in this figure.*

- Figure 5: It seems the annual mean temperatures are even more non-linear when considering the individual and combined forcings. Please discuss.

*The new figures reveal that the non-linearities have similar magnitudes.*

*Page 8, Line 11*

*"The difference between iL+oL and the sum of the hybrid experiments illustrate similar magnitude, but opposite differences for $T_{ann}$ and $T_{pw}$. The non-linear behavior here, in temperature as well as precipitation, is related to the varying response of the atmospheric circulation. The steep slopes of the ice sheet combined with katabatic winds and the anti-cyclonic circulation around the ice sheet margins generally limits the heat advection towards the interior ice sheet (Noël et al., 2014; Merz et al., 2016). Hence, potential precipitation and temperature changes on the interior ice sheet are largely dependent on changes in the circulation, which is not responding linearly to the combined insolation and oceanic forcings. The largest deviation is, however, found on the central west coast, where loss of snow cover and a strengthened albedo feedback explain the non-linearity (as described in relation to Fig. 3). "*

- Figure 6: remove 'iP+oP' from titles in line with the other figures.

*This has been done.*

Please also note the supplement to this comment:
http://www.clim-past-discuss.net/cp-2016-48/cp-2016-48-AC1-supplement.pdf

[Figure]

**Supplement:**

[revised manuscript text omitted]

---

## Author Comment (AC2) · 22 Aug 2016

**Authors' response to reviewer comments on "Greenland warming during the last interglacial: the relative importance of insolation and oceanic changes" by Rasmus A. Pedersen et al.**

We would like to thank the reviewers for the constructive comments. The authors' response to each comment is inserted below in *blue italics.*

Page and line numbers below mark the locations in the attached marked-up manuscript.

**Anonymous Referee # 2**

This manuscript evaluates the role of insulation and sea surface temperature changes on the Greenland temperature during the Eemian. This work is interesting and valuable as it could offer insights into (i) the drivers of sea level changes during that period, (ii) the drivers of climate change and (iii) the reasons for the discrepancy between modelled and reconstructed Greenland temperature. The work carried out is sound and well described (apart from a few minor clarifications that need to be made), but the implications of the results are not sufficiently well presented and some of the analysis needs to go a bit further. This paper could have a lot more impact with a little bit of adjustment to the manuscript and a little bit more analysis of the result. I therefore suggest the manuscript to be accepted after some corrections and clarification. These would be a bit more than minor revisions, but i don't anticipate they would require too much work.

Overall the manuscript does a good job at describing the changes associated with SST and insulation forcings, but does discuss the reasons of these changes. In particular, I would like to see some explanation of the role of insulation on precipitation seasonality.

*We have added more details to the discussion of the snowfall changes.*

*Page 7, Line 20*

*"During summer, iL+oP illustrates that the insolation contributes to the snowfall increase over the interior ice sheet. The fall pattern in iL+oL on the other hand indicates non-linear behavior, in that iL+oL does not resemble the sum of the two hybrid experiments: the increase on the eastern GrIS is only seen in iL+oL (as illustrated by difference in the bottom row of Fig. 4). Note, however, that Fig. 4 displays the relative change in snowfall, and the elevated northeastern region is very dry. The absolute snowfall anomaly (not shown) decreases rapidly towards the interior ice sheet, and the peak in the relative anomalies corresponds to a modest absolute increase (less than 0.2 mm/day). The non-linear behavior appears to be related to atmospheric circulation changes. The seasonal mean 10 m wind in iL+oL (not shown) has strengthened*

*southward flow along the northeast coast compared to the control climate and the hybrid experiments, which could contribute to increased orographic precipitation in the region; despite being mainly along to coast. The modest absolute precipitation increase in this region could, however, be related to very few storm events, that would not be evident from the seasonal mean circulation anomalies. "*

There is some mention of 'non-linearity' effect, but this is very much glanced over. It needs more description of what that means, how strong the non-linearity is and what causes it.

*We have added new plots to Figure 3-6 illustrating the difference iL+oL – (iL+oP + iP+oL). These have been used for more elaborate discussion of 'non-linearities' throughout the manuscript. Three examples are highlighted here:*

*Page 7, Line 3*

*"The largest difference is found in JJA near the Disko Bay on the central west coast, where the iL+oL warming is stronger than the sum of the hybrid experiments. As previously described, this region exhibits an albedo decrease due to loss of snow cover. The combination of snow melt driven by oceanic warming and the positive insolation anomaly in iL+oL gives rise to a strengthened albedo feedback that causes the apparent non-linearity. The insolation anomaly alone (in iL+oP) only causes a modest loss of snow cover, and the impact of the surface albedo feedback is therefore limited. "*

*Page 7, Line 21*

*"The fall pattern in iL+oL on the other hand indicates non-linear behavior, in that iL+oL does not resemble the sum of the two hybrid experiments: the increase on the eastern GrIS is only seen in iL+oL (as illustrated by difference in the bottom row of Fig. 4). Note, however, that Fig. 4 displays the relative change in snowfall, and the elevated northeastern region is very dry. The absolute snowfall anomaly (not shown) decreases rapidly towards the interior ice sheet, and the peak in the relative anomalies*

*corresponds to a modest absolute increase (less than 0.2 mm/day). The non-linear be-havior appears to be related to atmospheric circulation changes. The seasonal mean 10 m wind in iL+oL (not shown) has strengthened southward flow along the northeast coast compared to the control climate and the hybrid experiments, which could con-tribute to increased orographic precipitation in the region; despite being mainly along to coast. The modest absolute precipitation increase in this region could, however, be related to very few storm events, that would not be evident from the seasonal mean circulation anomalies. "*

*Page 8, Line 11*

*"The difference between iL+oL and the sum of the hybrid experiments illustrate similar magnitude, but opposite differences for $T_{ann}$ and $T_{pw}$. The non-linear behavior here, in temperature as well as precipitation, is related to the varying response of the atmo-spheric circulation. The steep slopes of the ice sheet combined with katabatic winds and the anti-cyclonic circulation around the ice sheet margins generally limits the heat advection towards the interior ice sheet (Noël et al., 2014; Merz et al., 2016). Hence, potential precipitation and temperature changes on the interior ice sheet are largely de-pendent on changes in the circulation, which is not responding linearly to the combined insolation and oceanic forcings. The largest deviation is, however, found on the central west coast, where loss of snow cover and a strengthened albedo feedback explain the non-linearity (as described in relation to Fig. 3). "*

In the discussion and introduction, clarify that part of the SST changes are caused by insulation and that this study focuses on the direct effect of insulation vs ocean temperature changes.

*We have rewritten part of the introduction to clarify this.*

*Page 3, Line 14*

*"Using a series of general circulation model (GCM) experiments, we assess the Green-*

[Figure]

*land climate during the Eemian. We investigate how the simulated changes could affect the GrIS surface mass balance and the ice core record, and compare the reconstructed and simulated temperatures. While the insolation change is the only forcing in our experiments, we further compare the direct impact of the insolation change and the indirect effect of retreating sea ice and increasing sea surface temperatures (SSTs). The direct and indirect impacts are separated using two hybrid experiments: one forced by Eemian insolation and fixed pre-industrial sea surface conditions (direct impact, "insolation-only") and one with pre-industrial insolation and Eemian sea surface conditions (indirect impact, "ocean-only"). The temperature change during the Eemian resembles that of future climate scenarios (e.g. Clark and Huybers, 2009), and our comparison could reveal whether the Eemian is an appropriate analogue for future climate change in Greenland; i.e. whether insolation or the ambient oceanic warming dominates the total response."*

Also, there should be a discussion of how well the model simulates modern Greenland temperature and how that would impact interpretation of the results. For example, some GCMs have difficulties simulating Arctic cloud processes. Could that affect the sensitivity of the model to changes in insulation/SSTs ?

*A comment on model performance has been added to the Methods section.*

*Page 4, Line 10*

*"Compared to the widely used version 2.3, which was included in the latest IPCC assessment report (Flato et al., 2013), the new EC-Earth version includes updated versions of both the atmosphere and ocean models. Comparison of a present-day simulation to gridded observational data reveals an improved overall performance in version 3.1 compared to the previous version with a few remaining biases (cf. Davini et al., 2014). Relevant for this study, the comparison reveals a cold bias over most of Greenland and a too extensive Arctic sea ice cover."*

The conclusions of the manuscript are a bit underwhelming. The start of the manuscript

suggests that this study could shed light on the reasons for model-data discrepancy regarding Greenland Eemian temperature. The paper concludes that changes in ice sheet topography are to be blamed, but that is precisely a factor that the paper was not including. Is there nothing to be learned about the model's sensitivity to insulation and SST changes ?

*The conclusion has been rewritten.*

*Page 10, Line 17*

[revised manuscript text omitted]

Finally the mass balance calculations are really interesting and valuable, but the results are a bit lost in the manuscript which is a real shame.

*We have elaborated on the SMB discussion through the manuscript. Examples from the abstract, the introduction and the conclusion:*

*Page 1, Line 12*

*"Surface mass balance calculations with an energy balance model further indicate that the combination of temperature and precipitation anomalies leads to potential mass loss in the north and southwestern parts of the ice sheet. The oceanic conditions favor increased accumulation in the southeast, while the insolation appears to be the dominant cause of the expected ice sheet reduction. Consequently, the Eemian is not a suitable analogue for future ice sheet changes. "*

*Page 3, Line 24*

*"In the assessment of the Greenland climate, we also aim to investigate how the simulated Eemian climate could impact the ice sheet. The ice sheet response is a combined result of dynamics (ice flow) and surface mass balance changes (melt and accumulation). Here, we employ a detailed surface scheme to assess the surface mass balance. Again, the assessment of the relative importance of the insolation and sea surface warming will indicate whether Eemian ice sheet reconstructions are useful analogues for future ice sheet changes."*

*Page 11, Line 26*

*"The combined impact of the simulated warming and snowfall increase could favor*

*substantial ice sheet changes. SMB calculations revealed that while the oceanic changes favor increased accumulation over the southeastern GrIS, the changed insolation causes increased melting along the coastal parts of the ice sheet. The hybrid experiments indicate that the insolation is the dominant factor behind the expected reduction of the GrIS. This reiterates the finding of van de Berg et al. (2011), that direct use of the relation between temperature and mass loss in the Eemian is likely to overestimate future greenhouse gas-driven melting. The SMB changes are consistent with previous ice sheet reconstructions (e.g. Born and Nisancioglu, 2012; Quiquet et al., 2013; Stone et al., 2013) suggesting ice sheet retreat in the southwest and northern coastal GrIS. "*

**Other minor comments**:

- Line 20: "While the ice core air content only suggests limited elevation changes at the NEEM site (45 ± 350 m higher than present ice sheet elevation), the NEEM ice core temperature reconstruction has been corrected using the surface elevation change estimate from the ice core air content (NEEM community members, 2013)." A lot of repetition in this sentence which I find a bit difficult to understand, so I suggest modifying it.

*The sentence has been rewritten.*

*Page 2, Line 24*

*"The NEEM ice core temperature reconstruction has been corrected for this effect using the ice core air content which suggests an elevation increase of 45±350 m relative to the present ice sheet elevation (NEEM community members, 2013)."*

- Section 2.3 page 2, line 20. Reference for the SST and sea ice boundary conditions. Is this from Pedersen et al. (2016b)?

*Correct; this has been clarified.*

*Page 5, Line 5*

[Figure]

*"We have designed four experiments to investigate how the last interglacial insolation changes impacted the climatic conditions on Greenland (cf. Table 1). An experiment with Eemian (125 ka) conditions ("iL+oL", full Eemian experiment) is compared to a pre-industrial control climate state ("iP+oP"). The simulations are forced with GHGs and insolation from the respective periods along with prescribed sea surface temperatures (SST) and sea ice concentration (SIC) obtained from fully coupled model experiments with identical GHGs and insolation (the coupled simulations are described in Pedersen et al., 2016b)."*

- Section 2.3 line 27: clarify, what the impact of insolation on SST changes is based on? is this again from Pedersen et al. (2016b) ?

*The sentence has been rewritten.*

*Page 5, Line 23*

*"In the coupled simulations from Pedersen et al. (2016b), the induced insolation forcing leads to sea ice retreat and increasing SSTs across high northern latitudes. Figure 2 depicts sea ice concentration and SST anomalies in the coupled simulations from Pedersen et al. (2016b), indicating the differences between the sea surface boundary conditions employed here. The sea ice reduction is primarily manifested as a northward retreat of the ice edge; the sea ice concentration in the central Arctic is largely unchanged. The strongest warming is found in the North Atlantic following the northward retreat of the sea ice edge and a strengthening of the Atlantic meridional overturning circulation (AMOC). The AMOC increase is related to a regional increase of the surface salinity and increased wintertime convection, which is in part related to biases in the pre-industrial control climate (see detailed description in Pedersen et al., 2016b)."*

- Page 5, line 29: "The simulated responses reveal that the ice sheet topography is important for the precipitation changes: Figure 4 reveals several examples of contrasting snowfall changes on the east and western side of the ice divide." I understand what is meant here, but I would suggest clarifying this statement as the readers may confuse

(i) the control that topography has on the pattern of climate change observed, with (ii) the effect of topographical changes not included here.

*This has been rewritten.*

*Page 7, Line 9*

*"Figure 4 reveals several examples of contrasting snowfall changes on the east and western side of the ice divide, illustrating the barrier effect of the ice sheet (e.g. Ohmura and Reeh, 1991). This pattern suggests that inclusion of ice sheet topography changes could impact the precipitation patterns; as illustrated by Merz et al. (2014b)."*

- Figure 6 add label for "effect of SST" "effect of insolation" above the subplots to help the reader understand the results.

*Explanatory labels have been added to all plots: "full", "insolation", and "ocean".*

- Page 6, line 31. This paragraph needs more discussion. The second sentence is not enough to justify the non-linearity. I suggest formalising slightly more the factor decomposition to calculate the interaction between ocean and insolation forcings (see Stein and Alpert) or at least state that adding the two effects does not give the full temperature change. Also, add a discussion of the reasons for this. Why is this non-linearity different for precipitation-weighted and absolute temperature difference? Can you explain the processes that lead to the non-linearity ?

Stein, U., Alpert, P., 1993. Factor Separation in Numerical Simulations. Journal of the Atmospheric Sciences 50, 2107–2115.

*We have added new plots to Figure 3-6 illustrating the difference iL+oL – (iL+oP + iP+oL). These have been used for more elaborate discussion of 'non-linearities' throughout the manuscript. Three examples are highlighted here:*

*Page 7, Line 3*

*"The largest difference is found in JJA near the Disko Bay on the central west coast,*

*where the iL+oL warming is stronger than the sum of the hybrid experiments. As pre-
viously described, this region exhibits an albedo decrease due to loss of snow cover.
The combination of snow melt driven by oceanic warming and the positive insolation
anomaly in iL+oL gives rise to a strengthened albedo feedback that causes the appar-
ent non-linearity. The insolation anomaly alone (in iL+oP) only causes a modest loss
of snow cover, and the impact of the surface albedo feedback is therefore limited. "*

*Page 7, Line 21*

*"The fall pattern in iL+oL on the other hand indicates non-linear behavior, in that iL+oL
does not resemble the sum of the two hybrid experiments: the increase on the east-
ern GrIS is only seen in iL+oL (as illustrated by difference in the bottom row of Fig.
4). Note, however, that Fig. 4 displays the relative change in snowfall, and the el-
evated northeastern region is very dry. The absolute snowfall anomaly (not shown)
decreases rapidly towards the interior ice sheet, and the peak in the relative anomalies
corresponds to a modest absolute increase (less than 0.2 mm/day). The non-linear be-
havior appears to be related to atmospheric circulation changes. The seasonal mean
10 m wind in iL+oL (not shown) has strengthened southward flow along the northeast
coast compared to the control climate and the hybrid experiments, which could con-
tribute to increased orographic precipitation in the region; despite being mainly along
to coast. The modest absolute precipitation increase in this region could, however, be
related to very few storm events, that would not be evident from the seasonal mean
circulation anomalies. "*

*Page 8, Line 11*

*"The difference between iL+oL and the sum of the hybrid experiments illustrate similar
magnitude, but opposite differences for $T_{ann}$ and $T_{pw}$. The non-linear behavior here,
in temperature as well as precipitation, is related to the varying response of the
atmospheric circulation. The steep slopes of the ice sheet combined with katabatic
winds and the anti-cyclonic circulation around the ice sheet margins generally limits*

[Figure]

*the heat advection towards the interior ice sheet (Noël et al., 2014; Merz et al., 2016). Hence, potential precipitation and temperature changes on the interior ice sheet are largely dependent on changes in the circulation, which is not responding linearly to the combined insolation and oceanic forcings. The largest deviation is, however, found on the central west coast, where loss of snow cover and a strengthened albedo feedback explain the non-linearity (as described in relation to Fig. 3). "*

Please also note the supplement to this comment:
http://www.clim-past-discuss.net/cp-2016-48/cp-2016-48-AC2-supplement.pdf

**Supplement:**

[revised manuscript text omitted]

---

## Author Response (AR1)

**Authors' response to editor decision and comments on *"Greenland during the last interglacial: the relative importance of insolation and oceanic changes"* by Rasmus A. Pedersen et al.**

**The authors' response to each comment is inserted below in *blue italics*.**

**Page and line numbers below mark the locations in the attached marked-up manuscript.**

**The marked-up manuscript is attached below the comments.**

**Editor Decision**: Publish subject to minor revisions (review by Editor) (26 Aug 2016) by Dr. Erin McClymont

Comments to the Author (pdf): cp-2016-48-comments-to-author.pdf

Comments to the Author:

The authors have made a detailed response to the comments of both reviewers, and have incorporated this feedback into a revised manuscript which is now stronger on emphasising the main arguments and findings of the paper, and its wider significance.

I have attached an annotated version of the author comment 1 where a few minor clarifications would still benefit the text - the comments are in sticky notes on the pdf. If these are not legible in the attachment I will provide a written summary.

**(1)    In addition, could the authors clarify whether the Table 2 caption should read "...change relative to iP+oP and associated..."**

*This typo has been corrected; it should be iP+oP as suggested.*

**(2)    Why 1000 years? is this to allow for certain flow patterns or feedbacks to be taken into account?**

Referring to **Page 9, line 8**
*»A rough estimate of the potential ice sheet height changes can be obtained by assuming that the simulated anomalies act over 1,000 years. In this view, the indicated anomalies on Fig. 6 can be read as "meters of water equivalent pr. 1,000 years".«*

*For this simple estimate, 1,000 years was chosen as the order-of-magnitude appropriate for ice sheet changes. We have updated the text to illustrate that 1,000 is only an example of a relevant time scale.*

***Page 9, line 8***

*»A rough estimate of the potential ice sheet height changes can be obtained by assuming that the simulated anomalies act over a relevant period; e.g. 1,000 years.«*

**(3)    If you say reduction and growth then I don't think you need the minus and plus symbols?**

Referring to **Page 9, line 10**

»Acting over 1,000 years, the extreme SMB values in Fig. 6 correspond to more than -550 m reduction near the southwest, northwest and northeastern coasts and more than 550 m growth near the southeastern coast.«

*The minus has been removed.*

**(4)    Add a line into the interpretations re-iterating this? could it account for some of the LIG NH autumn changes?**

Referring to **Reviewer #1 comment**:

*The results presented in this study have a strong focus on seasonal changes, thus it should be explained what the possible impact of using a fixed seasonal calendar could be.*

*The non-linearities arise even without any calendar effects. This can be illustrated e.g. by comparing the effects of the warmer SST in the two 'background climates' (PI + Eemian). Consider the two fields:*

*(1)  iP+oL – iP+oP = effect of Eemian SSTs (under PI insolation)*
*(2)  iL+oL – iL+oP = effect of Eemian SSTs (under Eemian insolation)*

*Note that (1) and (2) are unaffected by the choice of calendar as the compared simulations have identical insolation. (1) is depicted as the third row in Figure 4; (2) is equivalent to the difference between the anomalies in the first two rows. (1) has almost no snowfall increase in northeastern Greenland, while (2) has a substantial increase.*

**(5)    "...as the calendar we use is defined..."**

Referring to **Page 5, line 9**

»**As the calendar is defined** with a fixed vernal equinox, the largest difference between the two calendar definitions is found during northern hemisphere autumn.«

*This has been added.*

**(6)    Is this consistent with proxy data? especially when on line 31 you note that there are biases in the pre industrial control climate – we need to know if these simulations are realistic or if not, where not, since they have implications for the ice sheet results.**

Referring to **Page 5, line 15**

*»The sea ice reduction is primarily manifested as a northward retreat of the ice edge; the sea ice concentration in the central Arctic is largely unchanged. The strongest warming is found in the North Atlantic following the northward retreat of the sea ice edge and a strengthening of the Atlantic meridional overturning circulation (AMOC). The AMOC increase is related to a regional increase of the surface salinity and increased wintertime convection, which is in part related to biases in the pre-industrial control climate (see detailed description in Pedersen et al., 2016b). «*

*The related paper cited as Pedersen et al. (2016b) has now been published.*

> Pedersen RA, Langen PL, Vinther BM (2016) The last interglacial climate: comparing direct and indirect impacts of insolation changes. Clim Dyn. doi: 10.1007/s00382-016-3274-5

*Here, the coupled simulations are analyzed in detail and compared to proxy records of Eemian SST, sea ice cover, deep convection, and (tropical) precipitation. While we do not wish to reiterate the findings from this paper in too much detail, we have added an additional comment on the sea ice cover and the warming in the North Atlantic.*

**Page 5, line 18**

> *»The AMOC increase is related to a regional increase of the surface salinity and increased wintertime convection. The simulated pre-industrial climate has an extensive Arctic sea ice cover and a related lack of deep convection in the Labrador Sea. This contributes further to the Eemian North Atlantic warming which, despite regional agreement, is larger than suggested by proxy reconstructions in the central North Atlantic (see detailed description in Pedersen et al., 2016b).«*

(7)   **"The temperature anomalies generally resemble previous Eemian simulations (i.e. the multi-model mean from Lunt et al. 2013), but exhibit larger warming in the Arctic and the North Atlantic region."**

*- By how much?*

*We have specified this in the manuscript. Neither Lunt et al. (2013) nor IPCC AR5 (Masson-Delmotte et al. 2013) present an Arctic mean value. Hence, our comparison is based on spatial maps – making it impractical to present an accurate number.*

**Page 5, line 28**

> *»The temperature anomalies generally resemble previous Eemian simulations (i.e. the multi-model mean from Lunt et al. 2013), but exhibit larger warming in the Arctic and the North Atlantic region. The largest difference is found in the Nordic seas and the Labrador Sea, where our simulated annual mean warming is several degrees higher. Overall, the Arctic annual mean warming is less than 1 K higher than the multi-model mean. «*

**(8)** Title suggestion: "*Greenland temperature changes during the last interglacial: the relative importance of insolation and oceanic changes*".

*We prefer the shorter "Greenland during the last interglacial: the relative importance of insolation and oceanic changes". Reviewer #1 commented that the link between "warming" in the title and the SMB discussion in the abstract was unclear. We acknowledge this point, and as we discuss more than just the temperature changes, we prefer the more general title.*

**(9)** OK, but as the reviewer asks, are these changes over land, over ice, or some combination of both? Since you say on page 4 that you do not change the spatial extent of the ice sheet I interpret these changes as always over ice - but it would be useful to clarify this point in the text as the reviewer notes.

Referring to Page 6, line 1

»*During summer, strong warming patches are collocated with areas of albedo decrease due to loss of snow cover in coastal, low-elevation areas (not shown). The most prominent albedo change is found on the central west coast, south of Disko Bay, where the JJA mean surface albedo decreases by up to 0.4. Coastal areas in the northwest, central and northeastern regions exhibit albedo decreases of 0.1-0.2. The iP+oL simulation (ocean-only) shows similar but smaller magnitude JJA mean albedo changes, while iL+oP (insolation-only) has an almost unchanged JJA mean surface albedo. This difference illustrates that the increased shortwave absorption and subsequent larger sensible heat flux from the surface contributes to these local, near-coastal warming peaks in iL+oL and iP+oL.* «

*EC-Earth in the present version does not have an explicit glacier type. This means that if 'bare ice' is revealed on the ice sheet, the surface is allowed to warm beyond the freezing point. The surface elevation is fixed and the only difference between land and ice points is thus the albedo. The presented surface albedo anomalies reflect loss of snow cover with varying surface types underneath.*

**(10)** Is this related at all to the fixed season used in your experiments?

Referring to Page 7, line 2

»*The fall pattern in iL+oL on the other hand indicates non-linear behavior, in that iL+oL does not resemble the sum of the two hybrid experiments: the increase on the eastern GrIS is only seen in iL+oL (as illustrated by difference in the bottom row of Fig. 4).* «

***The answer to comment (4) applies here as well.***

*The non-linearities arise even without any calendar effects. This can be illustrated e.g. by comparing the effects of the warmer SST in the two 'background climates' (PI + Eemian). Consider the two fields:*

*(1) iP+oL – iP+oP = effect of Eemian SSTs (under PI insolation)*
*(2) iL+oL – iL+oP = effect of Eemian SSTs (under Eemian insolation)*

*Note that (1) and (2) are unaffected by the choice of calendar as the compared simulations have identical insolation. (1) is depicted as the third row in Figure 4; (2) is equivalent to the difference between the anomalies in the first two rows. (1) has almost no snowfall increase in northeastern Greenland, while (2) has a substantial increase.*

**(11)** **Correction: "…the coast."**

*This has been corrected.*

*Page 7, line 8*

[revised manuscript text omitted]